# Towards the quantized anomalous Hall effect in AlO$_x$-capped MnBi$_2$Te$_4$

Yongqian Wang[1,2,9], Bohan Fu[1,2,9], Yongchao Wang [3], Zichen Lian[3], Shuai Yang[1,2], Yaoxin Li [3], Liangcai Xu [3], Zhiting Gao[4], Xiaotian Yang[5], Wenbo Wang [5], Wanjun Jiang [3,6], Jinsong Zhang [3,6,7], Yayu Wang [3,6,7,8] & Chang Liu [1,2] ✉

The quantum anomalous Hall effect in layered antiferromagnet MnBi$_2$Te$_4$ harbors a rich interplay between magnetism and topology, holding a significant promise for low-power electronic devices and topological antiferromagnetic spintronics. In recent years, MnBi$_2$Te$_4$ has garnered considerable attention as the only known material to exhibit the antiferromagnetic quantum anomalous Hall effect. However, this field faces significant challenges as the quantization at zero magnetic field depending critically on fabricating high-quality devices. In this article, we introduce a straightforward yet effective method to mitigate the detrimental effect of the standard fabrication on MnBi$_2$Te$_4$ by depositing an AlO$_x$ layer on the surface before fabrication. Optical contrast and magnetotransport measurements on over 50 MnBi$_2$Te$_4$ demonstrate that AlO$_x$ can effectively preserve the pristine states of the devices. Surprisingly, we find this simple method can significantly enhance the anomalous Hall effect towards quantization, which resolves a longstanding challenge in the field of MnBi$_2$Te$_4$. Scaling relation analysis further reveals the intrinsic mechanism of anomalous Hall effect dominated by Berry curvature at various magnetic configuration. By tuning the gate voltage, we uncover a gate independent magnetism in odd-layer MnBi$_2$Te$_4$ devices. Our experiments not only pave the way for the fabrication of high-quality dissipationless transport devices, but also advance the investigation of exotic topological quantum phenomena in 2D materials.

Magnetic topological materials have emerged as a frontier in condensed matter physics, providing promising platforms for exploring exotic quantum phenomena and applications in topological spintronics[1–3]. For uncovering novel topological physics, successful fabrication of high-quality devices with quantized transport is prerequisite. As the first identified material possessing van der Waals characteristics, intrinsic magnetism, and nontrivial band topology simultaneously, MnBi$_2$Te$_4$ not only exhibits rich novel quantized phenomena when exfoliated down to few-layer limit[4–6], but is also considered capable of addressing the disorder issue that is prevalent in

[1]School of Physics, Renmin University of China, Beijing, China. [2]Key Laboratory of Quantum State Construction and Manipulation (Ministry of Education), Renmin University of China, Beijing, China. [3]State Key Laboratory of Low Dimensional Quantum Physics, Department of Physics, Tsinghua University, Beijing, China. [4]Beijing Academy of Quantum Information Sciences, Beijing, China. [5]School of Physical Science and Technology, ShanghaiTech Laboratory for Topological Physics, ShanghaiTech University, Shanghai, China. [6]Frontier Science Center for Quantum Information, Beijing, China. [7]Hefei National Laboratory, Hefei, China. [8]New Cornerstone Science Laboratory, Frontier Science Center for Quantum Information, Beijing, P. R. China. [9]These authors contributed equally: Yongqian Wang, Bohan Fu. ✉e-mail: liuchang_phy@ruc.edu.cn

magnetically doped topological insulators (TIs)[7]. The bulk crystal of MnBi$_2$Te$_4$ can be regarded as a stacking of Te–Bi–Te–Mn–Te–Bi–Te septuple layer (SL) along $z$-direction (Fig. 1a). A-type antiferromagnetic (AFM) structure with interlayer AFM order and intralayer ferromagnetic (FM) order forms below the Néel temperature ($T_N$) ~ 25 K. When interacting with band topology, this layer-dependent magnetic ordering can give rise to a rich variety of topological quantum states and exotic magnetoelectric response[8–15]. In odd-SL MnBi$_2$Te$_4$ film, the gapped Dirac topological surface states due to the parallel surface magnetizations drive the system into the quantum anomalous Hall (QAH) state with 1D dissipationless chiral edge state transport[8] (Fig. 1b). This manifests as a quantized Hall conductivity $\sigma_{xy} = Ce^2/h$ at zero-magnetic field ($\mu_0H = 0$), where $C$ represents the Chern number, $e$ is the electron charge, and $h$ is the Planck constant. In even-SL MnBi$_2$Te$_4$, the opposite surface magnetizations result in vanishing $\sigma_{xy}$ and lead to the axion insulator state characterized by a zero plateau at $\mu_0H$ (ref. 10). Recent progresses in MnBi$_2$Te$_4$ have unveiled a plethora of novel topological phenomena, including the Möbius insulator[16], layer Hall effect[11], axion optical induction[17], and quantum metric non-linear transport[18,19].

Although the QAH effect and axion insulator states have been observed in 5- and 6-SL MnBi$_2$Te$_4$, the temperature ($T$) below which quantization is realized remains much lower than its $T_N$. A more formidable challenge arises from the exceptionally low yield of MnBi$_2$Te$_4$ film exhibiting quantized transport. Over the past five years, neither perfectly quantized[9,12,13,15,20–22] nor zero plateau[12,17–19,22] at $\mu_0H = 0$ can be

consistently reproduced. The lack of quantization not only obstructs the discovery of new phenomena but also complicates the interpretation of available data. Possible reasons include various structural defects or impurity phases[23–27], instability of surface electronic structures[28–31], and weakened surface out-of-plane magnetic anisotropy (MA)[31–33]. Our recent studies combining optical contrast ($O_c$), transport, magneto-optical Kerr effect (MOKE) measurements revealed a substantial impact of fabrication on the properties of MnBi$_2$Te$_4$ devices[22]. The contact with photoresist not only reduces the $O_c$ value during the fabrication process, but may also results in mismatched even-odd-layer-dependent magnetotransport[22]. The mechanism likely originates from the formation of a dead insulating layer on the MnBi$_2$Te$_4$ surface, which is caused by the change of surface band structure[30,34,35]. Over the past five years, developing a low-damage fabrication method to reproduce the QAH effect has become one of the most pressing tasks in the field of magnetic topological quantum materials and devices.

In this work, we optimize the fabrication process by depositing an AlO$_x$ protective layer on MnBi$_2$Te$_4$ top surface prior to the standard electron beam lithography (EBL). To ensure a good ohmic contact between the MnBi$_2$Te$_4$ and electrode, we implement an additional Ar ion etching step to selectively remove the AlO$_x$ at the electrode areas after the lithography process. By employing this method, we can fabricate transport devices with good electrical contact while using an insulating AlO$_x$ layer to fully isolate the photoresist Polymethyl Methacrylate (PMMA). Through optical measurement on a series of

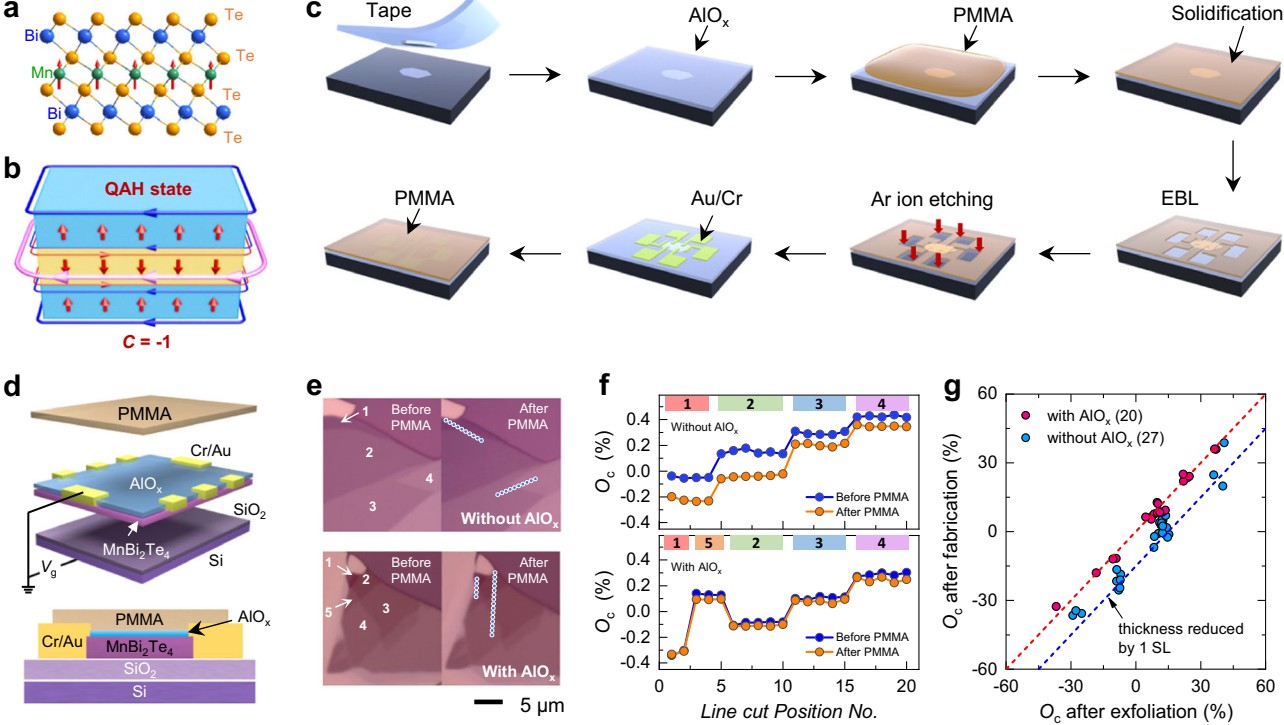

**Fig. 1 | Fabrication and optical contrast characterizations of few-layer MnBi$_2$Te$_4$ flakes. a** Crystal structure of MnBi$_2$Te$_4$. **b** Schematic of the QAH effect in an odd-SL MnBi$_2$Te$_4$. The red arrows in **a** and **b** represent the magnetic moments of Mn in each layer. **c** Illustration of the device fabrication process. This method is developed based on the standard EBL process. By simply depositing a thin layer of AlO$_x$ on the MnBi$_2$Te$_4$ surface, the PMMA resist is isolated from the top surface. The high insulation and compactness of AlO$_x$ make it possible to fabricate Hall bar patterns while protecting the sample from chemical reagents. **d** Front and side views of a transport device. **e** Optical images of MnBi$_2$Te$_4$ flakes exfoliated from the same crystal. The top (bottom) panel compares the color change of MnBi$_2$Te$_4$ flakes without (with) AlO$_x$ capping before and after contact with PMMA, respectively.

**f** Variation of $O_c$ for selected spots along the line traces in flakes of different thicknesses in (**e**). $O_c$ is defined as ($I_{flake}$ - $I_{substrate}$)/$I_{substrate}$, where $I_{flake}$ and $I_{substrate}$ are the intensity of MnBi$_2$Te$_4$ and substrate, respectively. **g** Statistical analysis of $O_c$ of 47 MnBi$_2$Te$_4$ flakes with (red) and without (blue) AlO$_x$ capping layer. Different data points represent the $O_c$ of different flakes after mechanical exfoliation and after contact with PMMA, respectively. The red and blue dashed line denote the $O_c$ reduction by 0 and 20%, respectively, corresponding to no change and a decrease of effective thickness by one layer. The variability of the degree of $O_c$ reduction among different flakes does not stem from measurement errors but rather arises from the inhomogeneities and sample quality fluctuations within the bulk crystal.

MnBi$_2$Te$_4$ thin flakes, we find that the fabrication issue induced by PMMA photoresist is largely mitigated. Most importantly, this simple idea overcomes the bottlenecks in the field of topological quantum materials over the past five years. We not only achieve the QAH effect in multiple MnBi$_2$Te$_4$ devices, but also reveal the key factors influencing the zero-magnetic-field quantization. Our work introduces a simple yet effective method for fabricating high-quality transport devices, paving the way for realizing the QAH effect and exploring more exotic topological quantum phenomena.

## Results

### Device fabrication and optical contrast

We got inspiration from previous experiments where those MnBi$_2$Te$_4$ devices exhibiting large anomalous Hall (AH) effect often had Al$_2$O$_3$ on the bottom of the flakes[8,21,36], either as a substrate or a supporting layer. This implies that the contact with Al$_2$O$_3$ may help to improve the quality of MnBi$_2$Te$_4$ device. Combined with our recent finding of the detrimental effect of fabrication on the top surface of MnBi$_2$Te$_4$, we come up with a straightforward yet effective idea that by depositing an AlO$_x$ layer on top of MnBi$_2$Te$_4$ to achieve the QAH effect. Figure 1c shows the schematic of the fabrication process. First, we transferred thick MnBi$_2$Te$_4$ flakes from a bulk crystal to the substrate using a Scotch tape. We then employed the mechanical exfoliation method to obtain the flakes with target thickness. The one-to-one correspondence between $O_c$ and thickness allows the rapid determination of SL number by optical method[11]. Subsequently, a 3-nm AlO$_x$ layer was deposited by thermal evaporation. We then adopted the standard EBL process to expose the designed Hall bar patterns. Next, the AlO$_x$ layer above the designed electrode areas was etched away by Ar ion etching, followed by the deposition of Cr/Au electrodes. To ensure the charge transport is not affected by the etching process in the contact region, we assessed the MnBi$_2$Te$_4$ thickness in the etched region by $O_c$ and atomic force microscopy measurements (see Supplementary Figs. 1 and 2). Finally, a PMMA layer was coated for further protection. The details of the fabrication are described in the Methods section. Compared to the Al$_2$O$_3$-assisted exfoliation and stencil mask method[8], our method is more straightforward and is based on the standard EBL process, which enables the fabrication of specific nano-devices with reduced sample size. Furthermore, since the AlO$_x$ is on the top surface, it not only offers an effective protection, but also can be used as a dielectric layer for top gate controllability.

Figure 1d depicts a schematic of a Hall bar device covered with AlO$_x$ capping layer and its cross-sectional view. To investigate the influence of AlO$_x$ on MnBi$_2$Te$_4$, we first compare the optical properties of MnBi$_2$Te$_4$ flakes with varied thicknesses, which were exfoliated from the same single crystal (Supplementary Fig. 3 for the $O_c$ change in each step). The optical images in Fig. 1e clearly suggest that for flakes without AlO$_x$ (up panel), the colors of all four regions change significantly before and after contact with PMMA. In contrast, those regimes with AlO$_x$ (down panel) do not exhibit noticeable change during the same process. To further explore the effect of AlO$_x$ quantitatively, we extract their $O_c$ values of the selected spots along the line traces in Fig. 1e and compare their variations directly. As shown in Fig. 1f, significant reductions of $O_c$ in all the four areas without AlO$_x$ are observed. In contrast, $O_c$ remains nearly unchanged for the five areas with AlO$_x$ capping layers. To eliminate the influences of device quality fluctuations on our observation, we compare the $O_c$ values of 47 MnBi$_2$Te$_4$ exfoliated from the same crystal (Fig. 1g). Different data points represent the $O_c$ of different flakes after exfoliation and after contact with PMMA. The distribution of $O_c$ falls well into two parts (red and blue dots). According to the one-to-one correspondence between $O_c$ and thickness[11], the decrease of $O_c$ indicates the reduction of effective thickness during fabrication. The red and blue dashed lines represent the $O_c$ reduction by 0 and 20 %, respectively, which corresponds to unchanged thickness and a decrease of thickness by one SL.

For the flakes without AlO$_x$ layer, regardless of their initial $O_c$ values, most of the samples display a pronounced decrease in $O_c$ after fabrication. Notably, the degree of $O_c$ change exhibits certain variability among different samples. This behavior does not stem from measurement errors but rather originates from the inhomogeneities and sample quality fluctuations within the MnBi$_2$Te$_4$ crystal. Such sample-dependent sensitivity to fabrication has also been observed in previous experiments[22]. These results clearly demonstrate that AlO$_x$ can effectively mitigates the damages caused by PMMA[22].

### Statistical analysis of the influence of AlO$_x$ on transport properties

In magnetic topological systems, the AH effect typically results from three mechanisms: intrinsic Berry curvature $\Omega(\mathbf{k})$, skew-scattering, and side-jump. In the transport of MnBi$_2$Te$_4$, due to defects or impurity phases, all the three mechanisms could contribute to the AH effect[20]. However, in an ideal quantized Hall system, the transverse transport should be dominated by $\Omega(\mathbf{k})$ in momentum ($\mathbf{k}$) space[37]. Theoretically, $\sigma_{xy}$ can be calculated by integrating $\Omega$ over $\mathbf{k}$, as expressed by:

$$\sigma_{xy} = -\frac{e^2}{2\pi h}\int \Omega(\mathbf{k})d^2\mathbf{k}$$

When the Fermi level ($E_F$) is tuned into the magnetic exchange gap, the integral of $\Omega$ equals the $C$ number multiplied by $2\pi$, resulting in the quantization of $\sigma_{xy}$ at $e^2/h$. To investigate the influence of AlO$_x$ on the intrinsic AH effect, we measured the transport properties of 17 odd-SL MnBi$_2$Te$_4$ devices. All the data shown in the main text was obtained at the charge neutral point (CNP) unless otherwise specified. Prior to this, we measured the current–voltage curve of the AlO$_x$ layer to exclude its contribution to transport (Supplementary Fig. 4). Figure 2a–d shows the $\mu_0 H$ dependence of $\sigma_{xy}$ and $\sigma_{xx}$ for two 7-SL devices exfoliated from the same thick flake on the same tape. Both devices show quantized $\sigma_{xy}$ at the high $\mu_0 H$ Chern insulator state ($C = -1$). However, their AH effects at zero field exhibit dramatically different behaviors. For device #1 without AlO$_x$ (Fig. 2a), $\sigma_{xy}$ almost vanishes at the low $\mu_0 H$ AFM regime. Such behavior is consistent with our previous observation that fabrication process can damage the top surface, leading to a reduction of effective thickness by one SL[22]. However, for the device #9 fabricated by the current method, a large $\sigma_{xy}$ accompanied with a square-shaped hysteresis is observed. The insets show the schematic distribution of the topological surface states wave functions (green) for devices without and with AlO$_x$. To further demonstrate the influence of AlO$_x$ on transport more clearly, we display the gate voltage ($V_g$) dependent $\sigma_{xy}$ and $\sigma_{xx}$ at $\mu_0 H = -8$ T and 0 for the two devices, as displayed in Fig. 2b, d. For device #1, $\sigma_{xy}$ smoothly crosses zero with increasing $V_g$, indicating a reduced gap size during the fabrication process[22]. In contrast, device #9 shows a wide plateau in $\sigma_{xy}$ in the same $V_g$ range of the Chern insulator, indicating an incipient QAH state in the AFM state.

For 2D materials, the transport properties of thin flakes are inevitably influenced by the fluctuations of device qualities. Previous experiments have suggested that MnBi$_2$Te$_4$ exhibits sample-dependent properties, even for flakes prepared from the same crystal[8,24,25]. Therefore, it is challenging to expect any fabrication method to guarantee the production of perfect QAH devices with 100 % certainty. To demonstrate that the enhancement of the AH effect arises from AlO$_x$, we studied the transport properties of 17 odd-number-SL MnBi$_2$Te$_4$ devices, with the data presented in Fig. 2e, f. These devices were numbered based on the increasing order of their $\sigma_{xy}$ at $\mu_0 H = 0$. Apart from devices #2, #4, #6, and #8 that were acquired from crystal #1, all the other 13 devices were obtained from crystal #2. All the devices exhibit $\sigma_{xy} = e^2/h$ at the high $\mu_0 H$ Chern insulator state, indicating the overall high quality of our devices. However, these devices show dramatically different behaviors in their AH effect.

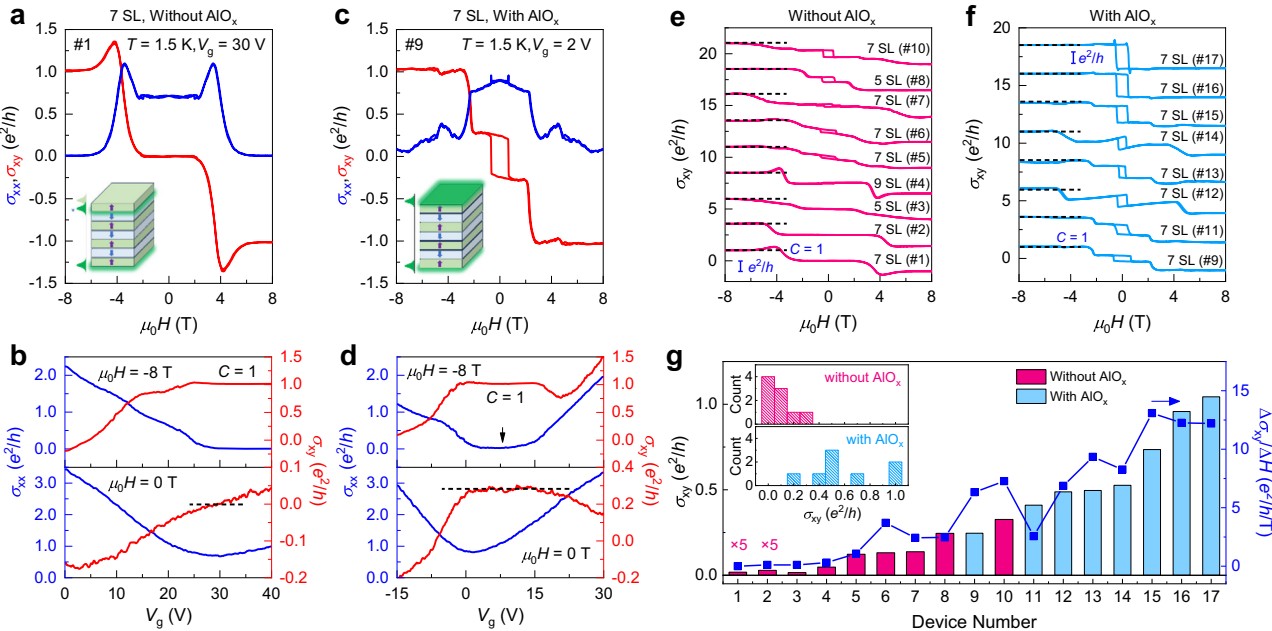

**Fig. 2 | Comparison of transport behaviors for devices obtained by different fabrication methods. a** Transport behaviors at CNP for a 7-SL device without AlO$_x$ covering layer. Due to fabrication effects, the surface state shifts down to the second SL (inset), and the hysteresis of $\sigma_{xy}$ near zero field almost disappears. **b** Variation of $\sigma_{xy}$ and $\sigma_{xx}$ as a function of $V_g$ at $\mu_0 H = -8$ T and 0, respectively. The black dashed line represents the $V_g$ window where $\sigma_{xy}$ plateaus coexist under zero and high $\mu_0 H$ conditions, respectively. **c** Transport behavior of a 7-SL device exfoliated from a MnBi$_2$Te$_4$ flake on the same tape, but with an AlO$_x$ layer deposited during the fabrication process. The large hysteresis indicates excellent protection of device performance. The inset illustrates that the topological surface state remains predominantly distributed on the outermost surface due to the protection of AlO$_x$. **d** In the same $V_g$ range of high field Chern insulator state (marked by black dashed line), $\sigma_{xy}$ at $\mu_0 H = 0$ exhibits a broad plateau during sweeping $V_g$. **e**, **f** $\mu_0 H$-dependent $\sigma_{xy}$ at $T = 1.5$ K for 17 odd-SL devices. The only difference during their fabrication lies in whether the surface was deposited with AlO$_x$. All the devices exhibit quantized $\sigma_{xy}$ at high $\mu_0 H$, as marked by the black dashed lines. **g** Summarized $\sigma_{xy}$ at $\mu_0 H = 0$ and $\Delta\sigma_{xy}/\Delta H$ values at the plateau transition for the 17 devices. Devices with AlO$_x$ capping layer (blue) generally show a larger AH effect than those without AlO$_x$ (red). The inset displays the histograms of AH effect distribution of the 17 devices. The size of each bin on the $\sigma_{xy}$-axis is 0.1 $e^2/h$.

Remarkably, the 9 devices without AlO$_x$ layer exhibit small $\sigma_{xy}$ at $\mu_0 H = 0$. And some devices even exhibit almost indiscernible hysteresis. In sharp contrast, all the other 8 devices with AlO$_x$ capping layer manifest large $\sigma_{xy}$ and square-shaped hysteresis, with two devices almost quantized at $e^2/h$. Figure 2g summarizes the $\sigma_{xy}$ at $\mu_0 H = 0$ of these devices. To largely avoid any artificial trend, we adopted the strategy in previous statistical studies of MnBi$_2$Te$_4$ crystals[25] by sorting the $\sigma_{xy}$ from smallest to largest. In the inset of Fig. 2g, we also present the histograms of their $\sigma_{xy}$ distribution. The size of each bin on the $\sigma_{xy}$-axis is 0.1 $e^2/h$. For the devices without AlO$_x$, their $\sigma_{xy}$ values are distributed within the range of 0 to 0.3 $e^2/h$. In contrast, for those devices with AlO$_x$, their $\sigma_{xy}$ distribution displays a clear shift towards higher values. Notably, despite not all devices with AlO$_x$ exhibit the QAH effect, their AH effects have already surpassed the values for most MnBi$_2$Te$_4$ devices in literatures[9,11–13,20,22]. In Fig. 2g, we also summarized the values of $\Delta\sigma_{xy}/\Delta H$ of the 17 devices (blue points), which represent the sharpness of magnetic transition. The variation of $\Delta\sigma_{xy}/\Delta H$ aligns with the trend of $\sigma_{xy}$. These results undoubtedly demonstrate that AlO$_x$ plays a significant role in enhancing the AH effect.

The fabrication of high-quality devices enables us to compare the influences of magnetic properties on transport. Figure 3a–c shows the $\mu_0 H$-dependent $\sigma_{xx}$ and $\sigma_{xy}$ for three devices obtained from the same crystal. Fortunately, for devices #11 and #16, they were obtained on the same substrate during one cleaving process. It enables us to further explore the influences of AlO$_x$ on MnBi$_2$Te$_4$ flake while preserving the consistency of the devices. Overall, the three devices manifest consistent transport behaviors, with their main differences being the values of $\sigma_{xx}$ and $\sigma_{xy}$ at $\mu_0 H = 0$. However, the sharpness of the plateau transition, which reflects the magnetic flipping process, differs dramatically. For device #11, the $\sigma_{xy}$ and $\sigma_{xx}$ at $\mu_0 H = 0$ are 0.5 and 1.1 $e^2/h$, respectively, and the transition is relatively gentle. For

device #16, although the value of $\sigma_{xx}$ does not change, $\sigma_{xy}$ is significantly improved to 0.96 $e^2/h$, comparable to the value in previous report[8]. In addition, the plateau transition is also sharper than that of device #11. Device #17 completely enters the QAH state, with $\sigma_{xy}$ reaching $e^2/h$ and $\sigma_{xx}$ dropping to 0. Because the QAH effect in magnetic TIs originates from the exchange field between local moments and electron spins[38]. The out-of-plane magnetic order plays a crucial role in the $\mu_0 H$-dependent transport behaviors. Therefore, the improved quantization along with the sharp $\sigma_{xy}$ transition suggests that device #17 is likely to have a stronger out-of-plane MA.

The scaling relation between $\sigma_{xy}$ and $\sigma_{xx}$ may further help us understand the role of AlO$_x$ in enhancing the QAH effect. Figure 3d–f displays the variation of $\sigma_{xy}$ as a function of $\sigma_{xx}$ during the cooling process under different $\mu_0 H$ and $V_g$s. As the AFM order strengthens at low $T$s, $\sigma_{xy}$ begins to exhibit behavior independent of $\sigma_{xx}$ and gradually approaches quantization, which is of typical the scaling behavior of the intrinsic AH effect dominated by $\Omega(\mathbf{k})$ (ref. 37). Upon increasing $\mu_0 H$, the device undergoes AFM, canted AFM, and finally enters FM state, accompanied by $\sigma_{xy}$ saturating at $e^2/h$ at higher $T$s. For device #11 with relatively weaker out-of-plane magnetic order, the exchange gap is expected to be smaller in the AFM state, thermal fluctuations could more easily smear out the role of $\Omega(\mathbf{k})$ (top in Fig. 3g). Therefore, complete quantization appears only when all moments are parallelly aligned because the gap is overall positively correlated with magnetization. However, for the device #17 with stronger out-of-plane order, the larger gap allows for $\sigma_{xy}$ reaching quantization even in the AFM state despite a small net moment. The influence of different magnetic configurations on exchange gap and $\Omega(\mathbf{k})$ is illustrated in Fig. 3g. $\Omega(\mathbf{k})$ can be interpreted as an effective $\mu_0 H$ in $\mathbf{k}$ space acting on electrons. The red and blue represent the distribution of $\Omega(\mathbf{k})$ of opposite sign in conduction and valence band, respectively.

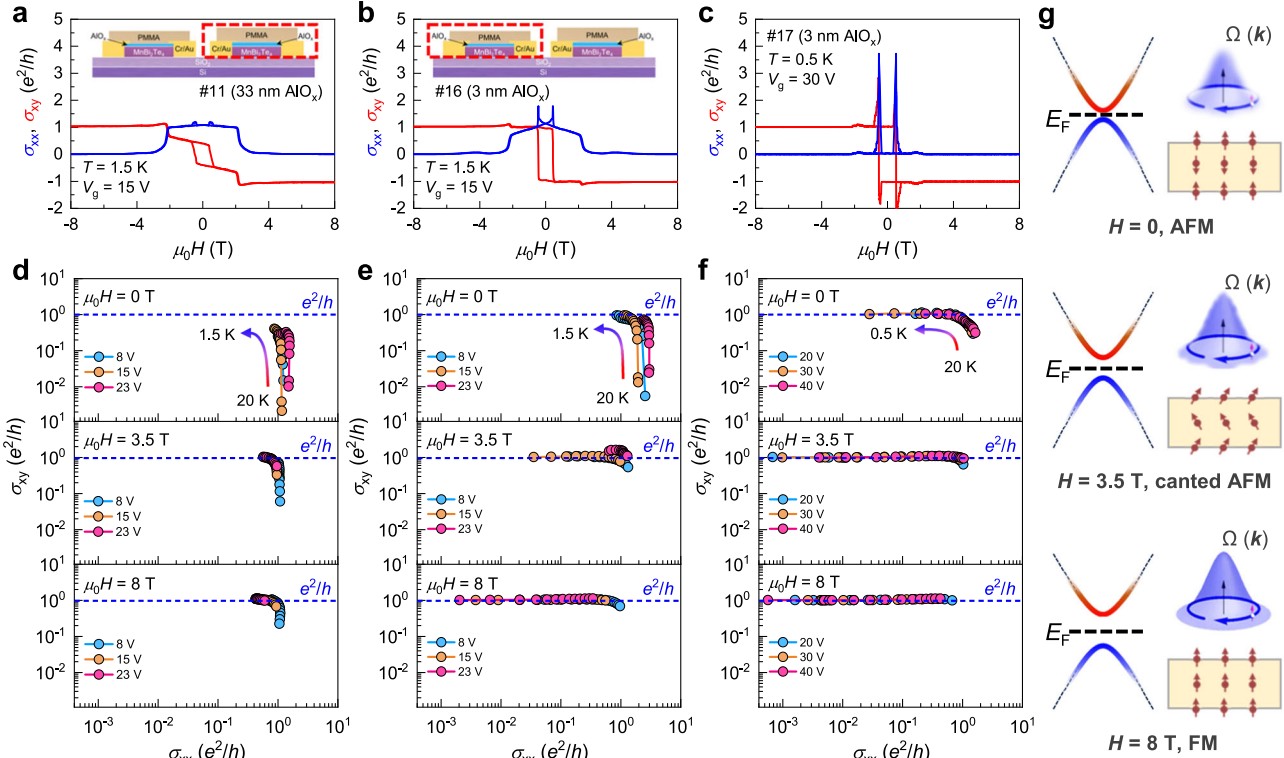

**Fig. 3 | Scaling relation between $\sigma_{xy}$ and $\sigma_{xx}$ of the intrinsic AH effect. a–c** $\mu_0 H$-dependent $\sigma_{xy}$ and $\sigma_{xx}$ for three 7-SL devices exfoliated from the same crystal. Devices #11 and #16 were obtained simultaneously in one cleaving process on the same substrate, with the former one undergoing an extra $AlO_x$ deposition process, having a thickness of 33 nm. **d–f** Evolution of $\sigma_{xy}$ with $\sigma_{xx}$ during the cooling process. With the formation of AFM order as lowering $T$s, the scaling relation between $\sigma_{xy}$ and $\sigma_{xx}$ at different $V_g$s gradually collapses into one single curve, and $\sigma_{xy}$ saturates at $e^2/h$. The $\sigma_{xx}$ independent behavior reflects the typical Berry curvature-dominated mechanism of the AH effect. **g** Schematics of the distribution of Berry curvature. From top to bottom, as the AFM order is tuned to the FM order, the out-of-plane component of the total magnetization is enhanced. As a result, the exchange gap increases, and the Berry curvature exhibits greater robustness against thermal fluctuations.

## Gate voltage-independent magnetism

Next, we investigate the influence of $V_g$ on the magnetic properties of $MnBi_2Te_4$ devices. Previous studies have revealed different $V_g$-dependent magnetism on magnetically doped TIs[39–41]. The electrical control of van der Waals magnetism has also attracted wide attention. As the first layered topological antiferromagnet, it remains unclear whether $V_g$ can exert similar effects. Figure 4a, b displays the $\mu_0 H$-dependent $\sigma_{xy}$ and $\sigma_{xx}$ for device #16 at various $T$s. The hysteresis vanishes at around $T = 21$ K, accompanied by the disappearance of $\sigma_{xx}$ peaks. To quantitatively investigate the changes in the AFM state, we extract the coercive field ($H_c$) values at different $V_g$s and plot them as a function of $T$ with an offset of 0.25 T (Fig. 4c). The $H_c$ dependence of $T$ can be well described by the power law $-(1 - T/T_N)^\beta$, where $\beta$ represents the critical exponent. We notice that $V_g$ has almost negligible effect on the AFM order. $T_N$ remains a constant at ~21.3 K and $\beta$ maintains at ~0.52. Similar results were also observed in previous neutron diffraction on $MnBi_2Te_4$ bulk crystal and reflectance magneto-circular dichroism (RMCD) measurement on exfoliated thin flakes[14,42]. Our experiments further point that this critical behavior cannot be tuned by a bottom $V_g$. Figure 4e shows the colormap of $\sigma_{xy}$ as a function of $V_g$ and $\mu_0 H$. It clearly shows that $H_c$ is independent of $V_g$, further supporting the $V_g$-independent magnetism in odd-SL $MnBi_2Te_4$ device. Reproducible results obtained from another two 7-SL $MnBi_2Te_4$ with and without $AlO_x$ capping layer are documented in Supplementary Figs. 5 and 6.

## Discussion

Finally, we discuss the possible mechanisms underlying the enhancement of AH effect. In our previous research, we found that the coating of PMMA during the EBL process reduces the $O_c$ of $MnBi_2Te_4$, leading to a reduction of effective thickness[22]. $AlO_x$ serves as an effective barrier by isolating the surface from direct contact with the resist, thus providing a substantial protection for $MnBi_2Te_4$. However, in the history of 2D materials, the most widely used and effective capping layer for protection is $h$-BN, rather than $AlO_x$. In fact, previous experiments have suggested that the oxidation process may alter the intrinsic properties of $MnBi_2Te$ film[28], therefore, using an $AlO_x$ capping layer to protect $MnBi_2Te$ is unconventional (Supplementary Fig. 7 for the aging effect). Furthermore, beyond employing a protective layer, a shadow mask method can also be employed to avoid direct contact with PMMA resist. However, many experiments have demonstrated that even employing these methods[13,18,20,21], the AH effect remains non-quantized. Interestingly, a comparison of recent transport experiments in $MnBi_2Te_4$ reveals that regardless of the different device preparation or electrode deposition methods, all $MnBi_2Te_4$ devices exhibiting pronounced AH effect have one surface in contact with $Al_2O_3$ (refs. 8,21,36). In these experiments, $Al_2O_3$ is positioned under $MnBi_2Te_4$ and does not provide any protection to the top surface. Hence, the simple protective role is insufficient to explain the close correlation between large AH effect and $AlO_x$ in current experiments. It naturally raises the question of whether $AlO_x$ may play an additional role beyond protection.

Our scaling relation studies imply that the enhancement of perpendicular magnetic order may be crucial for the QAH effect. Based on our experimental data and the results in previous studies, we discuss the potential additional roles that $AlO_x$ may play. A conceivable scenario is the electric field enhanced magnetism at the $AlO_x/MnBi_2Te_4$ interface[43,44]. However, this scenario can be largely excluded because our $V_g$-dependent experiments demonstrate that $V_g$ has a small

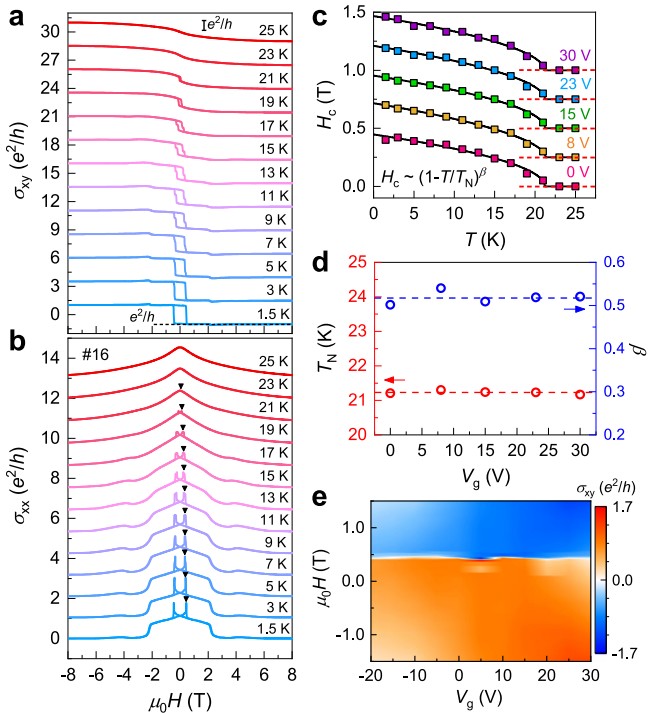

**Fig. 4 | Transport and magnetic properties tuned by $V_g$. a, b** $\mu_0 H$-dependent $\sigma_{xy}$ and $\sigma_{xx}$ at the CNP for device #16 at various $T$s. The hysteresis and the double peak in $\sigma_{xx}$ disappear at around $T = 21$ K. The black triangles mark the position of $H_c$ at different $T$s. **c** $H_c$ extracted from the field sweep data as a function of $T$ at varied $V_g$S. The solid squares are the data points. The black lines are the data fittings in the form of $-(1 - T/T_N)^\beta$. The red dashed lines represent the position of $H_c = 0$ for each curve. **d** Summarized $T_N$ and $\beta$ from the fittings as a function of $V_g$. The blue and red dashed lines represent the average positions of the $\beta$ and $T_N$ values for different $V_g$s. $T_N$ and $\beta$ are found to be -21.2 K and 0.52, respectively, both of which are independent of $V_g$. **e** Colormap of $\sigma_{xy}$ in the parameter space of $\mu_0 H$ and $V_g$. The boundary between blue and orange region marks the $V_g$-independent $H_c$.

influence on magnetism. Another possible scenario is the enhanced perpendicular magnetic anisotropy (PMA) by $AlO_x$. In spintronics, many experiments have demonstrated that depositing amorphous oxide (such as $AlO_x$, $MgO$, $TaO_x$, $HfO_x$) can substantially increase the interfacial PMA at the interface between oxides and magnetic materials[45–47]. Therefore, it is naturally expected that the $AlO_x$ layer strengthens the interfacial PMA of $MnBi_2Te_4$, which in turn enhances the AH effect. In fact, there has been theoretical calculations suggesting that the MA in monolayer $MnBi_2Te_4$ is weak due to the weak $p$-$d$ hybridization between Mn and Te (ref. 48). Later, inelastic neutron scattering pointed that for $MnBi_2Te_4$ crystals, the MA is enhanced by the interlayer two-ion anisotropy[32]. However, due to the absence of neighboring layers, the MA of the surface is still weak. These results naturally explain why, in all current $MnBi_2Te_4$ experiments exhibiting a large AH effect, the device must have at least one surface in contact with $AlO_x$.

To validate our conjectures, we conducted cryogenic magnetic force microscopy (MFM) measurement to directly visualize the magnetic properties across different regions of the same 7-SL $MnBi_2Te_4$. As anticipated, the region with $AlO_x$ manifests a much stronger magnetic signal compared to the region without $AlO_x$ (Supplementary Fig. 8). Moreover, to further explore the influence of $AlO_x$ on QAH effect, we compared the magnetic hysteresis loops of two fully quantized devices with single-sided and double-sided $AlO_x$ contacts. Interestingly, the device with both top and bottom surfaces in contact with $AlO_x$ shows a larger $H_c$ (see Supplementary Fig. 9 for details). In magnetic materials, $H_c$ is proportional to the strength of PMA[49]. The larger $H_c$ in device with

double-sided $AlO_x$ aligns with the finding that $AlO_x$ can enhance the interfacial PMA[45–47]. In addition to PMA, a recent calculation has also suggested that bringing $MnBi_2Te_4$ surface close to a polar insulator can modify the surface potential, which is helpful for the QAH effect[30]. As a polar insulator, $Al_2O_3$ may also play an additional role in enhancing the QAH effect. It is worth noting that although all current experiments support the scenario that $AlO_x$ likely contributes to magnetism, the microscopic mechanism remains inadequately understood owing to the challenges in directly measuring the interfacial magnetism. Further studies are required to elucidate the exact mechanisms.

In summary, we report the successful realization of the QAH effect in $MnBi_2Te_4$ devices capped with $AlO_x$ by employing a revised fabrication method based on the standard EBL. By simply depositing an $AlO_x$ layer on top of $MnBi_2Te_4$, we observe a substantial enhancement of the AH effect, ultimately achieving quantization. Our experiments resolve a longstanding challenge in the field of magnetic topological materials, paving the way for fabricating high-quality devices and investigating the intricate interplay between nontrivial topology and 2D magnetism. Recently, novel transport phenomena unavailable in previous QAH systems have already been observed in 7-SL $MnBi_2Te_4$ with current configuration, and the enhancement of surface magnetism by $AlO_x$ is considered crucial to explaining these new phenomena[50]. This simple yet effective method is not only significant for fundamental studies, but also lays the groundwork for creating novel topological spintronic devices[3,43,47]. It is important to note that the current exploration on utilizing $AlO_x$ to achieve the QAH effect is still in the initial stage. More refined control of the $AlO_x$ growth parameters[45,46] and the interface of oxide/$MnBi_2Te_4$ would further optimize the QAH effect, which remains a promising topic for future studies.

## Methods
### Crystal growth
High-quality $MnBi_2Te_4$ single crystals were synthesized by directly mixing $Bi_2Te_3$ and MnTe with a ratio of 1:1 in a vacuum-sealed silica ampoule. For crystal #1, the mixture was first heated up to 700 °C, and then slowly cooled down to 591 °C, followed by a long period of annealing process. The phase and crystal structure were examined by X-ray diffraction on a PANalytical Empyrean diffractometer with Cu Kα radiation. For crystal #2, a small amount of Te was added to the mixture, with the ratio between $Bi_2Te_3$, MnTe, and Te modified to 1:1:0.2. The ampoule was then slowly heated to 900 °C and maintained at this temperature for 1 h. Subsequently, it was cooled down to 700 °C, holding for 1 hour and then gradually cooled to 585 °C and maintained for 12 days. After the annealing, the ampoule was quenched in water to avoid phase impurities. Apart from devices #2, #4, #6, and #8, all other 13 devices were obtained from crystal #2. The four devices have already represented the samples showing the largest AH conductivity in the devices prepared from crystal #1.

### Device fabrication
$MnBi_2Te_4$ flakes were mechanically exfoliated onto 285 nm thick $SiO_2$/Si substrate by using the Scotch tape method in an Ar-filled glovebox with $O_2$ and $H_2O$ level lower than 0.1 ppm. Initially, the substrate was thoroughly cleaned with acetone, isopropanol, and deionized water. Then the surface of $SiO_2$/Si was treated with air plasma at -125 Pa for 3 min. The tape-covered substrate was heated up to 60 °C for 3 min to facilitate smooth exfoliation of the single crystals into flakes. Micrometer-sized thin flakes can be obtained by mechanically exfoliation on thick flakes for several times. The thickness was identified by optical contrast measurement in the glovebox immediately after exfoliation. After the target flakes were obtained, a 3-nm aluminum was deposited onto the surface using a thermal evaporator with a deposition rate 0.04 nm/s under a vacuum better than $4 \times 10^{-4}$ Pa. Oxygen was then introduced into the chamber, and the aluminum layer was oxidized for 5 min at a pressure of $2 \times 10^{-2}$ Pa. For device #11, an extra deposition process with

longer time was employed to compare the influence of different $AlO_x$ parameter on transport. During this process, an additional 30-nm $AlO_x$ was deposited under a controlled oxygen environment at a pressure of $2 \times 10^{-2}$ Pa.

To assess the effect of PMMA on the $MnBi_2Te_4$ samples, 270 nm thick PMMA was spin-coated onto the samples in an Ar-filled glovebox at a controlled speed of 4000 round/min. The samples were then heated at 60 °C for 7 min and left to stabilize in the glovebox for 24 h. Subsequently, the samples were than immersed in acetone for 20 min, rinsed with acetone followed by isopropanol, and their optical contrasts were measured immediately after the removal of PMMA. Standard EBL was employed on $MnBi_2Te_4$ samples to pattern the Hall bar structure. The oxidized aluminum was first etched from the sample surface using an Ar ion milling machine at a pressure of $2 \times 10^{-4}$ Torr for 75 s. Cr/Au electrodes (3/50 nm) were then deposited using a thermal evaporator connected to a glovebox. Following this, the samples were again spin-coated with PMMA adopting the same parameters as before for further protection.

### Transport measurement

Standard four probe transport measurements for devices #1 to #16 were carried out in a cryostat with the lowest $T$ of -1.5 K and an out-of-plane magnetic field up to -8 T. The longitudinal and Hall voltages were acquired simultaneously via two lock-in amplifiers with an AC current (100 nA, 13 Hz) generated by a Keithley 6221 current source meter. For device #17, the transport was performed in a dilution refrigerator with AC current excitation of 10 nA at 13 Hz. To correct the geometrical misalignment, both the longitudinal and Hall signals were symmetrized and antisymmetrized with respect to the magnetic field. The back-gate voltage was applied by a Keithley 2400 source meter through the $SiO_2$/Si substrate.

### MFM measurement

Cryogenic MFM experiments were conducted in a commercial atomic force microscope (atto-AFM) equipped with commercial cantilevers (spring constant $k \approx 2.8$ N/m and resonance frequency $\approx 75.8$ kHz) in a closed-cycle helium cryostat. An out-of-plane magnetic field was applied using a superconducting magnet. MFM images were taken in a constant height mode with lift height of -200 nm. The MFM signal, the change of cantilever resonance frequency, is proportional to the gradient of out-of-plane stray field. Electrostatic interaction was minimized by balancing the tip-surface potential difference.

## Data availability

All data supporting the finding in the study are presented within the main text and the supplementary information. All data are available from the corresponding author upon request.

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

## Acknowledgements

The authors appreciate the assistance provided by Prof. Yang Wu and Dr. Hao Li during the MnBi2Te4 crystal growth. Chang Liu was sponsored by National Natural Science Foundation of China (Grant No. 12274453), Beijing Nova Program (Grant No. 20240484574), and Open Research Fund Program of the State Key Laboratory of Low-Dimensional Quantum Physics (Grant No. KF202204). Jinsong Zhang was supported by National Natural Science Foundation of China (Grants Nos. 12274252 and 12350404). Yayu Wang was supported by the Basic Science Center Project of Natural Science Foundation of China (Grant No. 52388201) and the New Cornerstone Science Foundation through the New Cornerstone Investigator Program and the XPLORER PRIZE. Yayu Wang, Jinsong Zhang, and Chang Liu acknowledge the support from Innovation Program for Quantum Science and Technology (Grant No. 2021ZD0302502). Wenbo Wang was sponsored by National Key Research and Development Program of China (Grant No. 2022YFA1403000), and National Natural Science Foundation of China (Grant No. 12374161).

## Author contributions

C.L. conceived the project. C.L., Y.Y.W., J.S.Z., W.J.J., and W.B.W. supervised the research. Y.C.W. grew the MnBi2Te4 crystals, Y.Q.W., B.H.F., Z.C.L., and S.Y. fabricated the devices and performed the transport measurements with the help of Y.C.W., Y.X.L., L.C.X., and Z.T.G. X.T.Y. and W.B.W. performed the MFM measurements. C.L. and Y.Q.W. prepared the manuscript with comments from all authors.

## Competing interests

The authors declare no competing interests.
