## [Transparent Peer Review file · Nature Communications]

Towards the quantized anomalous Hall effect in AlOx-capped MnBi₂Te₄

Corresponding Author: Professor Chang Liu

Editorial Note: Parts of this Peer Review File have been redacted as indicated to maintain confidentiality regarding private communications.

Version 0:

Reviewer comments:

Reviewer #1

(Remarks to the Author)

The authors studied the effect of AlOx capping on the fabrication and transport properties of odd layer MnBi₂Te₄ devices. They found that the AlOx capping not only addresses the degradation issue due to the use of PMMA layer by producing more consistent thickness behavior, it also significantly improves the quality of zero-field transport properties. Close to quantized R_{xy} values were observed in two devices at zero field, with one of them fully entering the quantum anomalous Hall regime. They interpret the possible mechanism of the AlOx capping as enhancing the interface perpendicular magnetic anisotropy that stabilizes the out-of-plane magnetic order in the outmost MBT SL. The realization of zero-field quantization in MBT devices is highly desired, yet the results in the field have been inconsistent. The results from this paper provide a potential solution that is straightforward to implement which could significantly benefit this field. I would recommend its publication in Nature Communications. I do have a few questions for the authors listed below.

Ar ion etching is used to remove the AlOx and expose the MBT areas for making electrical contacts. Could the MBT layers also get etched in this process? If so, it might create regions of thinner MBT than the bulk of the device. Since topological transport is highly sensitive to edge conduction, have the authors done any characterization to make sure that the observed transport is not affected by the contact region?

I am a bit confused by the reduction of optical contrast without AlOx capping. In this case, does the physical thickness change, e.g., the PMMA somehow removes the entire top layer, or does it only chemically modify the top layer such that it becomes presumably insulating? If it is the latter case, the dead layer is still on the top of the active MBT layers. Then why would the optical contrast reduce to that in a thinner layer? The dead layer should still produce an additional optical contrast.

Reviewer #2

(Remarks to the Author)

In this manuscript, the authors report on a statistical study on MnBi₂Te₄ (MBT) devices with and without an AlOx capping layer, to improve the zero field quantized Hall response, which is expected to emerge for odd number SL MBT devices. Up to now, reproducing the zero field quantized Hall response for odd SL MBT devices has been challenging, and we acknowledge the importance of device optimization for the quantum anomalous Hall effect. We appreciate the extensive study of the authors, who show by analyzing 50 MBT samples with optical contrast measurements and 17 MBT devices with transport measurements, a clear relation between an improved anomalous Hall response and the capping with AlOx. Although we are impressed with the amount of work and believe the experimental results collected are valid, we do not fully corroborate the author's interpretation and presentation of the data, which prevents us from recommending the manuscript for publication in Nature Communication. Please see the points below.

1) The authors demonstrate that an AlOx capping layer enhances the quality of odd SL MBT devices, leading to higher anomalous Hall resistance and better temperature scaling. However, this result is not particularly surprising, as it is well-known that protecting MBT generally results in higher-quality devices. The authors claim the novelty of

their work lies in the unique role of the AlOx capping material in achieving the QAH effect. However, their arguments for this unique role are not entirely convincing.

The main argument presented by the authors involves comparing their method to other methods (references 13, 18, 32, 34). This comparison is not entirely valid due to numerous external factors influencing the realization of a quantized Hall resistance, such as crystal quality and fabrication details. Additionally, the cited studies do not offer as many statistical data points as this manuscript. Given the limited data points from studies using other fabrication methods and the significant variation in anomalous Hall resistance for devices with AlOx capping (Fig.2g, even with AlOx capping, the R_{xy} varies between $0.25G_0$ and G_0), the other methods may also be possible to demonstrate a 'better' quantization if more samples were fabricated and it is difficult to conclude that AlOx uniquely influences the material beyond providing protection.

2) The authors present a statistical study on MBT devices, but the data presentation is inadequate. Specifically, figure 2g is misleading, as it suggests an artificial trend with increasing device numbers. This is problematic and can misrepresent the findings. A more accurate approach would be to present the data as a distribution or histogram, with σ_{xy} on the x-axis and counts on the y-axis. Although creating a proper histogram might be challenging due to the large number of devices required, this method would more accurately reflect the paper's observations. However, even without such statistical representation of the data, one can already see that while AlOx improves σ_{xy} , there is still a significant spread among AlOx-capped devices (from $0.25G_0$ to G_0 , more than 75%), indicating that a perfectly quantized device may be more of an exception than the rule.

3) In the discussion section, the authors speculate about the possible physical mechanisms behind the improved anomalous Hall resistance due to AlOx. While the observation of improved resistance is evident, the manuscript lacks new physical evidence to support any of the proposed mechanisms. The authors present temperature scaling for three AlOx-capped devices, suggesting improved magnetic order. However, again, there is a significant spread among the AlOx-capped devices, and no direct physical relationship to the AlOx-capping layer is established.

Reviewer #3

(Remarks to the Author)

This paper by Wang et al. presents a systematic (and heroic) study of a large number of MnBiTe4 devices by examining their optical contrast and magnetotransport. The main conclusion is that introducing an AlOx layer atop the MBT can preserve the quality of the flake through the fabrication process, leading to an improvement of the anomalous Hall effect.

As mentioned by the authors, deterministically fabricating high-quality MBT devices has been a key roadblock for further exploration of topological quantum phenomena in MBT. This paper demonstrates a new fabrication procedure that can mitigate this important issue, which is supported by data from many samples. Therefore, I recommend publication in Nature Communications. Below are some comments for further improving the manuscript.

1. It is unclear to me what the key technical advances are when reading the last paragraph of the introduction (line 80 on page 3). It would be useful to more clearly contrast how the new method differs from the previous methods.
2. It's hard to see the difference between before and after images in Fig. 1e. It may be better to show the line traces of the O_c signal before/after for flakes with and without AlOx. Such a plot would be more transparent than Fig. 1f.
3. This study has not realized a deterministic fabrication of QAH effect. The authors should thus replace "resolve" to mitigate" in the abstract (line 33).
4. It might be useful to generate a plot similar to Fig. 2g but for the coercive fields and/or the sharpness of the transition.
5. It might be helpful to compare similar data as in Fig4 for a lower-quality device without AlOx. Such a comparison would strengthen the authors' claim.

Minor points:

- The acronym AH is not defined.
- Add references to "neither perfectly quantized [...] can be consistently reproduced." line 69 on page 3

Reviewer #4

(Remarks to the Author)

Version 1:

Reviewer comments:

Reviewer #1

(Remarks to the Author)

My previous questions have been addressed by the authors. I would recommend publication.

Reviewer #2

(Remarks to the Author)

The authors have addressed all the questions with detailed and thoughtful responses. They have carefully revised both the text and the figures, incorporating our feedback. The addition of an MFM experiment not only enhances the clarity of the research but also demonstrates the authors' rigorous approach. I am now satisfied with the revised manuscript and fully support its publication.

Reviewer #4

(Remarks to the Author)

In this letter we provide a point-to-point response to the reviewers' comments.

In the following, the reviewers' original comments are shown by blue italic characters.

The original sentences in the reference are indicated by red italic characters.

The authors' responses are shown by black normal characters.

Reviewer #1:

The authors studied the effect of AlO_x capping on the fabrication and transport properties of odd layer MnBi_2Te_4 devices. They found that the AlO_x capping not only addresses the degradation issue due to the use of PMMA layer by producing more consistent thickness behavior; it also significantly improves the quality of zero-field transport properties. Close to quantized R_{xy} values were observed in two devices at zero field, with one of them fully entering the quantum anomalous Hall regime. They interpret the possible mechanism of the AlO_x capping as enhancing the interface perpendicular magnetic anisotropy that stabilizes the out-of-plane magnetic order in the outmost MBT SL. The realization of zero-field quantization in MBT devices is highly desired, yet the results in the field have been inconsistent. The results from this paper provide a potential solution that is straightforward to implement which could significantly benefit this field. I would recommend its publication in Nature Communications. I do have a few questions for the authors listed below.

We thank the reviewer for the nice summary and the favorable comments of our work.

1. Ar ion etching is used to remove the AlO_x and expose the MBT areas for making electrical contacts. Could the MBT layers also get etched in this process? If so, it might create regions of thinner MBT than the bulk of the device. Since topological transport is highly sensitive to edge conduction, have the authors done any characterization to make sure that the observed transport is not affected by the contact region?

We thank the reviewer for raising this important point. The reviewer is correct that the etching process can potentially reduce the thickness of the MBT layer at the electrode positions, as we also

noticed this effect during our initial exploration of the best etching parameters. However, through careful optimization and selection of appropriate parameters, we can largely mitigate this issue in our experiment. Using the parameters shown in our manuscript, we did not observe any significant changes in the MBT thickness. To ensure the integrity of these areas, we employed optical contrast (O_c) and atomic force microscopy measurements to characterize the thickness of the MBT layer in the contact region before and after the etching process. These measurements confirm that the MBT layer in the contact regions remains unaffected by the etching procedure.

Firstly, we can assess the layer thickness through color change. Figure R1 shows a comparison of optical images of two MBT samples that were etched under different parameters. The AlO_x on the MBT surface within the blue frame was subjected to the Ar ion etching process, while the region outside the blue frame remained unetched. Notably, for the optical image obtained using the etching parameters in our manuscript (Fig. R1a), there is no discernible variation in the O_c , suggesting that the MBT thickness in the contact region remains largely unaffected. In contrast, when the etching parameters deviate away from the optimal values, such as increased etching duration, a substantial reduction in O_c is observed, as displayed in Fig. R1b. The reduction of O_c is indicative of a decrease in the MBT thickness when inappropriate parameters are employed. Based on the O_c measurement, our etching process does not affect the thickness of the contact regions, and therefore, the transport behavior will not be affected.

Fig. R1. Optical images of MBT flakes etched using different parameters. **a**, Optical image of MBT using the optical parameters shown in our manuscript. **b**, Optical image of MBT with longer etching time. The blue frame marks the etched region.

To more quantitatively identify the impact of etching on the MBT layer, we performed atomic force microscopy measurements on an etched MBT device. The main results are shown in Fig. R2. The atomic force microscopy measurements revealed that the step height at the edge of the etched regime (Fig. R2b) precisely corresponds to the thickness of the deposited AlO_x capping layer (Fig. R2a). Within the resolution of the atomic force microscopy, we did not detect significant reduction in the MBT thickness. The atomic force microscopy measurements, along with the variation of O_c , demonstrate the effectiveness of our carefully tuned etching process in preserving the integrity of the MBT layer.

Fig. R2. Atomic force microscopy measurement results of the thickness of AlO_x and the step height of the etched region. **a**, Morphology of AlO_x and height profile along the red line. The thickness of AlO_x is about 3 nm. **b**, Morphology and step height near the etched area. The height at the boundary of the etched regime is approximately 3 nm, consistent with the thickness of the AlO_x layer.

Furthermore, we would like to point out that even if the etching process slightly reduces the thickness of the MBT beneath the electrodes, it may not significantly affect the transport properties. This is because the MBT layer at the electrode locations is well covered by Cr/Au electrodes, which ensures that the entire electrode area remains equipotential. The key feature of chiral edge states in

the QAH or Chern insulator state is their topological robustness, meaning that imperfections in the boundary structure do not affect charge transport. In fact, all our samples require using a needle to remove the thick flakes around the target flake before making Hall bar structures. This process will lead to rough boundaries with much greater thickness non-uniformity than the changes induced by Ar etching process, as shown by the peak behavior in the atomic force microscopy result (Fig. R3a). However, our scanning microwave impedance microscope (MIM) measurements clearly show that such imperfections of edge structure do not affect the existence of edge states, and the MBT beneath the Au electrodes and other edge locations remain at equipotential (as shown in Figs. R3b and R3c). Therefore, we believe that even if the etching process slightly changes the thickness of MBT, its impact on the transport of chiral edge states is likely minimal.

Fig. R3. Atomic force microscopy and scanning MIM measurement results of the scratched region. **a**, Atomic force microscopy measurement result of the etched region of an MBT device. The height near sample edge is much higher than that of the bulk. **b-c**, Optical image and scanning MIM results of another MBT fabricated through the same process. Despite imperfections in the boundary, the edge states are clearly observed near the sample edge. Reprinted Figs. R3b and R3c with permission from Lin, W., Feng, Y., Wang, Y. et al. Influence of the dissipative topological edge state on quantized transport in MnBi_2Te_4 . *Phys. Rev. B* 105, 165411 (2022). Copyright (2022) by the American Physical Society.

The impact of contact area MBT thickness on the edge state transport is an intriguing direction that warrants further investigation in future studies. Following the reviewer's suggestion, we have incorporated some discussions in the revised manuscript to address the possible effect of Ar etching

on the MBT thickness. We have also included the relevant experimental data in the supplementary information to provide a more comprehensive understanding of this issue.

2. I am a bit confused by the reduction of optical contrast without AlO_x capping. In this case, does the physical thickness change, e.g., the PMMA somehow removes the entire top layer, or does it only chemically modify the top layer such that it becomes presumably insulating? If it is the latter case, the dead layer is still on the top of the active MBT layers. Then why would the optical contrast reduce to that in a thinner layer? The dead layer should still produce an additional optical contrast.

We are grateful to the reviewer for highlighting this issue. The effect of PMMA on O_c has been a central focus of our research, as understanding its underlying mechanism and addressing the issue is of great importance. Despite the fact that its physical mechanism has not yet been fully elucidated, our optical and magnetic measurements have yielded some valuable information.

The reviewer points that if the reduction in O_c is caused by a dead layer, then “*the dead layer is still on the top of the active MBT layers*”. We concur with this assessment and acknowledge that a decrease in O_c does not necessarily suggest a reduction in the physical thickness. To gain a deeper understanding of this phenomenon, we conducted a series of measurements in which we measured the physical thickness, magneto-optical Kerr effect (MOKE), and phonon frequency of a few MBT flakes with varied O_c before and after the PMMA contact [see supplementary information of *Nature Communications* 15, 3399 (2024)]. The key findings are presented in Fig. R4. Our results revealed that the physical thickness identified by atomic force microscopy measurement (Figs. R4e and R4f) and the coherent interlayer phonon frequency identified by Raman spectroscopy measurement (Fig. R4i) remained largely unchanged after the contact with PMMA, suggesting that PMMA does not remove the top layer of MBT. However, MOKE measurements indicated a significant weakening of the magnetic properties. These results led us to conclude that the reduction in O_c is primarily an indication of a decrease in “effective thickness” rather than a decrease in “physical thickness”.

Fig. R4. Optical images, atomic force microscopy, MOKE, and phonon frequency measurements of MBT before and after PMMA contact. **a-b**, Optical images of MBT after exfoliation and after PMMA contact. **c-d**, Atomic force microscopy images of the areas labelled by red boxes. **e-f**, Cross-sectional profiles of MBT along the red lines in **c** and **d**. **g-i**, Optical images, MOKE, and phonon frequency for a MBT before and after PMMA contact. Figures are adapted from the supplementary information of *Nature Communications* 15, 3399 (2024).

Regarding the reviewer’s question, “*Why would the optical contrast reduce to that in a thinner layer*”, we believe this phenomenon likely stems from modifications in the electronic structures of the material. In the realm of 2D materials, the principle behind using O_c to determine thickness is rooted in the process of light reflection and refraction at different interfaces. Therefore, the optical properties of each layer play a crucial role in the determination of thickness. Calculations based on the Fresnel’s law from classical optics can establish a one-to-one correspondence between O_c and thickness (or layer number) [*Nano Lett.*, 7(9), 2758 (2007)]. It is well-established that the optical properties of a material are intimately linked to its electronic structure. Therefore, any modification in the band structure of the surface during the fabrication is likely to influence the light transmission through this layer, as well as the refraction and reflection at its interface with the underlying layers

and vacuum. This leads us to the explanation that PMMA “*only chemically modifies the top layer such that it becomes presumably insulating*”. It is well known that insulators are usually transparent, thus an insulating dead surface layer may not contribute to O_c . Therefore, the change of O_c can be interpreted as a reduction in the effective thickness relevant to charge transport.

Interestingly, we have indeed noticed some experimental work proposed possible mechanisms to explain changes in MBT surface. For instance, the synergistic influence of a high concentration of Mn-Bi mixings and Te vacancies can trigger a surface reconstruction process, transforming an ideal MnBi_2Te_4 SL into a QL of Mn-doped Bi_2Te_3 and a bilayer of $\text{Mn}_x\text{Bi}_y\text{Te}$ [*Acs Nano*, 14, 11262 (2020)]. The fabrication process may further catalyze the instability of the surface. In this situation, the physical thickness remains unchanged but the surface band structure is modified. This naturally explains the reduction of effective thickness.

We believe that the new fabrication method presented in our manuscript will provide valuable experimental insights into the fabrication issues. In the revised manuscript, we have expanded our discussion on the reduction of effective thickness and incorporated relevant references that explore the potential mechanisms behind this phenomenon.

Reviewer #2:

In this manuscript, the authors report on a statistical study on MnBi_2Te_4 (MBT) devices with and without an AlO_x capping layer, to improve the zero field quantized Hall response, which is expected to emerge for odd number SL MBT devices. Up to now, reproducing the zero field quantized Hall response for odd SL MBT devices has been challenging, and we acknowledge the importance of device optimization for the quantum anomalous Hall effect. We appreciate the extensive study of the authors, who show by analyzing 50 MBT samples with optical contrast measurements and 17 MBT devices with transport measurements, a clear relation between an improved anomalous Hall response and the capping with AlO_x . Although we are impressed with the amount of work and believe the experimental results collected are valid, we do not fully

corroborate the author's interpretation and presentation of the data, which prevents us from recommending the manuscript for publication in Nature Communication. Please see the points below.

We thank the reviewer for the nice summary of our work and the favorable comment that “*we acknowledge the importance of device optimization for the quantum anomalous Hall effect.*” and “*we appreciate the extensive study of the authors, who show by analyzing 50 MBT samples with optical contrast measurements and 17 MBT devices with transport measurements, a clear relation between an improved anomalous Hall response and the capping with AlO_x.*”. The main concerns raised by the reviewer pertain to the interpretation and presentation of our data. We are very grateful for the constructive suggestions provided by the reviewer, which are very helpful in enhancing the quality of our work.

Following the reviewer's suggestions, we have reorganized our data presentation and revised the discussion part on the unique role of AlO_x. To support our data interpretation, we have included our latest results of magnetic force microscopy (MFM) measurements in the revised supplementary information. Our new experiments clearly suggest the enhanced magnetism in MBT flake with the AlO_x capping layer. Additionally, the increased coercive field (H_c) observed in another MBT device with double-sided AlO_x contact, which also manifests the perfect quantum anomalous Hall (QAH) effect, provides further evidence that AlO_x may play a role beyond simple protection. The transport data was also included in the supplementary information. We hope that the revised manuscript now meets the reviewer's standard for publication in *Nature Communications*.

1. The authors demonstrate that an AlO_x capping layer enhances the quality of odd SL MBT devices, leading to higher anomalous Hall resistance and better temperature scaling. However, this result is not particularly surprising, as it is well-known that protecting MBT generally results in higher-quality devices. The authors claim the novelty of their work lies in the unique role of the AlO_x capping material in achieving the QAH effect. However, their arguments for this unique role are not entirely convincing.

We are grateful to the reviewer for the comments and concur with the assertion that “*it is well-known that protecting MBT generally results in higher-quality devices.*”. However, we would like to emphasize that despite the widespread recognition of the importance of sample protection in the MBT community over the past five years, no group has ever successfully developed a specific and effective method to achieve this goal. Moreover, there has been a lack of theoretical or experimental evidence suggesting that covering MBT with AlO_x can give rise to MBT devices with higher quality. Our group has made a progress in this field by not only identifying the most significant factor that affects MBT device quality [*Nature Communications* 15, 3399 (2024)], but also proposing a simple and effective solution in the current manuscript. Our approach not only demonstrates the protective role of AlO_x, but also enables us to replicate the QAH effect, a challenging target that has not been accomplished since 2020 [*Science* 367, 895-900 (2020)].

We fully agree with the reviewer regarding the significant role of protection in enhancing the QAH effect. However, we respectfully disagree with the notion that “*this result is not particularly surprising*”. A careful examination of the history of van der Waals materials revealed that the most widely used and effective capping layer for protection is *h*-BN, rather than AlO_x. In fact, previous experiments have shown that the oxidation process can alter the intrinsic properties of MBT [*Adv. Funct. Mater.* 32, 2202234 (2022)]. Therefore, using surface-deposited AlO_x to protect MBT is not a common approach. If we review previous experimental studies on the transport of MBT, we will find that all those top groups have tried *h*-BN to protect MBT [*Nano Lett.* 21, 2544 (2021) & *Nature* 595, 521 (2021) & *Nat. Commun.* 13, 1668 (2022) & *Nat. Commun.* 13, 6191 (2022)]. Some of the studies have even employed the same crystals and fabrication method as those adopted in our work. Despite extensive efforts by numerous groups, including our own, none were able to reproduce the QAH effect until the AlO_x approach was proposed. The fact that *h*-BN, the commonly recognized best protective material, was unable to facilitate realizing the QAH effect, while the less commonly used AlO_x succeeded in doing so, naturally raises the question of whether AlO_x may play additional roles beyond mere protection.

The reviewer mentioned that “*The authors claim the novelty of their work lies in the unique role of the AlO_x capping material in achieving the QAH effect. However, their arguments for this unique role are not entirely convincing*”. We apologize for not clarifying this point. It is important to note that we did not intend to emphasize the unique role of AlO_x as the novelty of our work. Just as we wrote in the abstract that “*In this article, we introduce a straightforward yet effective method to resolve the detrimental effects of the standard fabrication on MnBi_2Te_4 by depositing an AlO_x layer on the surface before fabrication.*” The proposal of a new fabrication method, which provides effective sample protection and facilitates the reproduction of the QAH effect, represents the main novelty of our work. This is also the most significant contribution of our work to the field of MBT and topological states of matter.

The main argument presented by the authors involves comparing their method to other methods (references 13, 18, 32, 34). This comparison is not entirely valid due to numerous external factors influencing the realization of a quantized Hall resistance, such as crystal quality and fabrication details. Additionally, the cited studies do not offer as many statistical data points as this manuscript. Given the limited data points from studies using other fabrication methods and the significant variation in anomalous Hall resistance for devices with AlO_x capping (Fig.2g, even with AlO_x capping, the R_{xy} varies between $0.25G_0$ and G_0), the other methods may also be possible to demonstrate a ‘better’ quantization if more samples were fabricated and it is difficult to conclude that AlO_x uniquely influences the material beyond providing protection.

The reviewer mentioned that “*the cited studies do not offer as many statistical data points as this manuscript.*” and “*the other methods may also be possible to demonstrate a better quantization if more samples were fabricated*”. We appreciate the reviewer’s comments. However, it is important to clarify that the absence of statistical comparisons in the literature is not necessarily due to a lack of experimental data. Instead, it is likely that after numerous attempts with different single crystals and fabrication methods failed to achieve quantization, presenting a statistical study did not provide significant additional insights. [REDACTED] While we agree with the reviewer that “*the other methods may also be possible to demonstrate a better quantization if more samples were fabricated.*”

However, at least up to now, our AlO_x method remains the most effective and straightforward approach for achieving the QAH effect in MBT.

Furthermore, we wish to emphasize that the sample quality in our studies is already the highest available to date. In prior experiments, not only was the quantization at zero fields challenging, but also achieving quantization in magnetic fields was difficult. In fact, the success rate of fabricating devices with high-field quantization is quite low, with only a few groups worldwide capable of this feat. In our study, all the 17 devices exhibit good quantization under magnetic fields, which clearly indicates that we have minimized most of extrinsic factors at the crystal level. To further minimize the effects of confounding variables, such as “*crystal quality and fabrication details*” as mentioned by the reviewer, we have taken great care to adopt the same single crystal and flake throughout our experiments and to ensure consistency during each step of the fabrication process. By adopting this, we aim to isolate the influence of AlO_x as the primary factor affecting the QAH effect. Just as the reviewer commented, there is “*a clear relation between an improved anomalous Hall response and the capping with AlO_x* .” Regarding the fluctuation of their AH effect, it is important to recognize that such variability is a common challenge in all 2D materials studies, particularly after subjecting them to complex exfoliation and fabrication process [*Nature* 602, 41 (2022)]. It is well-known that even within the same crystal, fluctuation can arise when the bulk material is exfoliated to thin flakes [*Science* 367, 895-900 (2020) & *PRB* 104, 115168 (2021) & *NPJ Quantum Materials* 7, 7 (2022)]. This is why we invested considerable efforts in fabricating many devices and conducting statistical analyses to demonstrate the effect of AlO_x .

In response to the reviewer’s comments, we have revised and reorganized our discussion part on the role of AlO_x in the revised manuscript. We emphasized that the proposal of a new fabrication method represents the main novelty of our work. We also toned down our statement regarding AlO_x playing additional unique roles, instead presenting it as a possibility that cannot be ruled out based on current experiments. To verify our conjecture on the potential role of AlO_x , we have added our new results from MFM measurements on MBT with and without AlO_x , and transport measurements on MBT with single-sided and double-sided AlO_x contact. These new experiments provide valuable

physical insights into the role of the AlO_x in enhancing the QAH effect (see Figs. R6 and R7 in the response to question 3).

2. The authors present a statistical study on MBT devices, but the data presentation is inadequate. Specifically, figure 2g is misleading, as it suggests an artificial trend with increasing device numbers. This is problematic and can misrepresent the findings. A more accurate approach would be to present the data as a distribution or histogram, with σ_{xy} on the x-axis and counts on the y-axis. Although creating a proper histogram might be challenging due to the large number of devices required, this method would more accurately reflect the paper's observations. However, even without such statistical representation of the data, one can already see that while AlO_x improves σ_{xy} , there is still a significant spread among AlO_x -capped devices (from $0.25G_0$ to G_0 , more than 75%), indicating that a perfectly quantized device may be more of an exception than the rule.

We appreciate the reviewer's insightful comments and acknowledge that our data presentation may be misleading. Following the suggestion from both reviewers #2 and #3, and adopting the data presentation format from an earlier statistical study on the gap of MBT crystals (Fig. R5a), we have undertaken a revision of Fig. 2g. The revised figure is presented in Fig. R5b. The histogram of AH effect distribution with σ_{xy} on the x-axis and counts on the y-axis for the 17 devices are displayed in the inset. To eliminate any artificial trend, we adopted the strategy in previous statistical studies (Fig. R5a) by sorting the σ_{xy} for the 17 devices from smallest to largest. It provides a more accurate representation of the enhancement of the AH effect by AlO_x . Following the suggestion of reviewer #3, we have also summarized the $\Delta\sigma_{xy}/\Delta H$ values for the 17 devices (blue points), which represents the sharpness of the magnetic transition. The trend of the sharpness of the transition aligns with the trend of the enhancement of the AH effect by AlO_x .

Fig. R5. Sample dependent behaviors in MnBi_2Te_4 studies. **a**, Dirac point gap sizes of 15 MnBi_2Te_4 samples. **b**, Summarized σ_{xy} values at $\mu_0 H = 0$ for the 17 devices. Devices with AlO_x capping layer (blue) show a larger σ_{xy} than the devices without AlO_x (red). The inset in **b** presents the histograms of the AH effect distribution for 17 devices. Reprinted Fig. R5a from Shikin, A. M., Estyunin, D. A., Zaitsev, N. L. et al, Sample-dependent Dirac-point gap in MnBi_2Te_4 and its response to applied surface charge: A combined photoemission and ab initio study. *Phys. Rev. B* 104, 115168 (2021). Copyright (2021) by the American Physical Society.

The reviewer mentioned “*there is still a significant spread among AlO_x -capped devices (from $0.25G_0$ to G_0 , more than 75%), indicating that a perfectly quantized device may be more of an exception than the rule.*”. We respectfully disagree with the reviewer on this point. It is important to recognize that sample-dependent behavior is a pervasive phenomenon in the experimental study of MBT, as evidenced by both transport and spectroscopic measurements. It has been demonstrated that different MBT crystals, and even different thin flakes exfoliated from the same single crystal, can manifest different behaviors [*Science* 367, 895-900 (2020) & *PRB* 104, 115168 (2021) & *NPJ Quantum Materials* 7, 7 (2022)]. Considering this inherent variability, it is unrealistic to expect any

fabrication method to guarantee the production of perfectly quantized devices with 100 % certainty. Therefore, the fluctuation of σ_{xy} does not necessarily indicate that a perfectly QAH device is more of an exception than the rule. We should evaluate the effect of AlO_x from a statistical perspective. Our main objective is to propose a low-damage approach for fabricating high-quality MBT devices from a fabrication perspective, and to increase the probability of obtaining device with QAH effect. We would like to emphasize that even considering the fluctuation, the AH values of most our AlO_x -capped devices have surpassed the highest value reported by other groups, except for the quantized data by Yuanbo Zhang's group. To further demonstrate the effectiveness of our method, we recently fabricated another MBT device with both surfaces in contact with AlO_x , and observed better QAH effect (see Fig. R7).

Following the reviewers' suggestion, we have revised Fig. 2g and included the histograms of AH effect distribution for 17 devices with σ_{xy} on the x -axis and counts on the y -axis. The fluctuation behaviors of the AH effect were also discussed in the revised manuscript.

3. In the discussion section, the authors speculate about the possible physical mechanisms behind the improved anomalous Hall resistance due to AlO_x . While the observation of improved resistance is evident, the manuscript lacks new physical evidence to support any of the proposed mechanisms. The authors present temperature scaling for three AlO_x -capped devices, suggesting improved magnetic order. However, again, there is a significant spread among the AlO_x -capped devices, and no direct physical relationship to the AlO_x -capping layer is established.

We are grateful to the reviewer for highlighting this crucial point. In fact, the ability of AlO_x to enhance the interfacial magnetism has been extensively studied in the field of magnetic materials. A wealth of experimental evidence has demonstrated that depositing an AlO_x layer on the surface of a magnetic material can lead to an enhancement of interfacial perpendicular magnetic anisotropy. This phenomenon is not limited to AlO_x , but has been noticed with a wide range of amorphous and crystalline oxides, as emphasized by Dieny and Chshiev in this review article [*Rev. Mod. Phys.* 89, 025008 (2017)]. The authors explicitly state that “*Actually, this perpendicular magnetic anisotropy*

was found to be very common at magnetic metal/oxide interfaces since it has been observed with a large variety of amorphous or crystalline oxides, including AlO_x , MgO , TaO_x , HfO_x ". Our findings are consistent with the expectations from these previous experiments. Given the trend that devices with AlO_x capping layer always tend to exhibit a larger AH effect, it is a logically natural to consider that AlO_x may play a role in enhancing the magnetism of MBT.

Fig. R6. MFM measurements on two different regions with and without AlO_x capping layer in the same 7-SL MBT flake. The images were obtained at their coercive field H_c when magnetic contrast signals are the most pronounced. The blue and red colors represent the direction of downward and upward total magnetization.

We acknowledge that our speculation in the original manuscript lacked sufficient experimental evidence. In the past three months, we have been dedicated to conducting new experiments to gain more insight into the role of AlO_x . Below we present our preliminary imaging and transport results. We conducted MFM and transport measurements on several MBT thin flakes with different device configuration, to explore whether AlO_x plays a role on magnetism. Although MFM cannot directly measure the absolute magnetization, the magnetic contrast signal at the spin flipping magnetic field (coercive field) allows us to identify the relative strength of the magnetism. Similar technique has already been employed in our previous studies on the ferromagnetism of Cr/V-doped $(\text{Bi,Sb})_2\text{Te}_3$ QAH systems [*Nat. Phys.* 14, 791 (2018)]. Remarkably, our new MFM results align well with our

expectations. We first exfoliated a 7-SL MBT flake using the mechanical cleavage technique. Then, a part of the flake was covered with PDMS. An AlO_x capping layer was then deposited over the entire surface. After that, we removed the PDMS and employed a sharp needle to divide the sample into separate regions with and without AlO_x . This process enables the direct comparison of the effect of AlO_x on magnetism in the same sample. Figure R6 displays the MFM results obtained from two regions of the same flake. It clearly suggests that the region with AlO_x exhibits a stronger magnetic contrast signal than that without AlO_x . At the coercive field in which the magnetic configuration flips from $\downarrow\uparrow\downarrow\uparrow\downarrow$ (blue) to $\uparrow\downarrow\uparrow\downarrow\uparrow$ (red), a pronounced contrast signal is observed in the AlO_x capped part. Here \downarrow and \uparrow represents the down and up magnetization of each layer. Conversely, the magnetic contrast signal is significantly weaker in the region without AlO_x . The diminished magnetism even hinders the stable presence of downward magnetization in the domain flipping region, making it difficult to observe a distinct contrast signal (blue). Notably, in previous studies, even in MBT flake cleaved in ultra-high vacuum, the expected exchange gap was not observed [PRX9, 041038 (2019)]. It was suggested that *the ordered spin state in the bulk state may show fragility at the surface layers*. Our imaging result aligns with this observation.

In addition to MFM, we have also conducted transport measurements to explore the properties of MBT flake with both surfaces in contact with AlO_x . Before mechanical exfoliation, we deposited a 3-nm AlO_x on the bottom surface of MBT. Then, we cleaved the crystal onto a Si/SiO₂ substrate and deposited an AlO_x capping layer on the top surface using the previously established parameters. If AlO_x indeed enhances the interfacial perpendicular magnetic anisotropy, as suggested in previous studies [Rev. Mod. Phys. 89, 025008 (2017)], we would expect that contacting both top and bottom surfaces with AlO_x would further enhance the anisotropy. In conventional magnetic materials, the coercive field (H_c) is generally proportional to the perpendicular magnetic anisotropy [Coey, J. M. D. *Magnetism and Magnetic Materials*. Cambridge University, 2010]. Therefore, by comparing H_c values of devices with single-sided and double-sided AlO_x contact, we can infer the influence of AlO_x on the magnetic properties of MBT. As shown in Fig. R7, our transport results are consistent with this expectation. For MBT device with double-sided AlO_x contact, we observed a significant

enhancement in H_c compared to the device with single-sided AlO_x contact. Again, this observation supports our speculation that AlO_x may contribute to the enhancement of magnetism.

Fig. R7. The QAH effect in two MnBi_2Te_4 flakes with different AlO_x configurations. Both samples exhibit full quantization at zero magnetic field.

In addition to the comparative experiments, our recent transport measurements have unveiled a series of novel phenomena arising from layer dependent magnetic transition [*arXiv*: 2405.08686]. To understand the new phenomena, we have performed theoretical calculations based on a modified AFM spin chain model that considers the potential influences of AlO_x on the surface magnetization. Interestingly, we find that only by considering the enhanced surface magnetization can we explain the entire set of experimental findings.

In concluding our response, we would like to highlight an interesting observation from recent transport studies on QAH effect in MBT [*Science* 367, 895 (2020) & *Natl. Sci. Rev.*, 11, nwad189 (2023) & *arXiv*: 2401.11450]. Despite the diversity in device preparation and fabrication methods, such as mechanical exfoliation or molecular beam epitaxy, and the fabrication of Hall bar structure using either EBL or shadow masks, a common thread emerges among the samples that have shown zero field quantization: all MBT samples have one surface in contact with AlO_x . This commonality is particularly intriguing in the work done by Qi-Kun Xue group and Yuanbo Zhang group, where AlO_x is located underneath the MBT, serving no apparent protective role. The consistent presence

of AlO_x in enhancing the QAH effect, regardless of its position, again implies that AlO_x may play additional roles beyond mere protection in facilitating the realization of the QAH effect.

Following the reviewer's suggestions, we have moderated our statements regarding the unique role of AlO_x in the revised manuscript for greater accuracy. We have also included our preliminary MFM and transport results in the supplementary information, which provide physical evidence for the potential additional role of AlO_x . We are actively pursuing further studies and hope to provide more evidence in the future. We believe these revisions have improved the accuracy and readability of the manuscript, and hope it meets the standard for publication in *Nature Communications*.

Reviewer #3:

This paper by Wang et al. presents a systematic (and heroic) study of a large number of MnBi_2Te_4 devices by examining their optical contrast and magnetotransport. The main conclusion is that introducing an AlO_x layer atop the MBT can preserve the quality of the flake through the fabrication process, leading to an improvement of the anomalous Hall effect.

*As mentioned by the authors, deterministically fabricating high-quality MBT devices has been a key roadblock for further exploration of topological quantum phenomena in MBT. This paper demonstrates a new fabrication procedure that can mitigate this important issue, which is supported by data from many samples. Therefore, I recommend publication in *Nature Communications*. Below are some comments for further improving the manuscript.*

We appreciate the reviewer's nice summary and recommendation of our work.

1. It is unclear to me what the key technical advances are when reading the last paragraph of the introduction (line 80 on page 3). It would be useful to more clearly contrast how the new method differs from the previous methods.

We thank the reviewer for the constructive suggestion. In the previous preparation of transport devices, Hall bar structures were patterned using EBL, which involved exposing the PMMA resist

on the MBT surface. However, our previous work demonstrated that direct contact between PMMA and the sample could damage the surface, leading to a reduction in the effective thickness for charge transport [*Nature Communications* 15, 3399 (2024)]. Therefore, in our current study, we introduced an insulating AlO_x layer on the sample surface to isolate MBT from PMMA. For transport devices, it is essential that the surface remains free of insulating layers before depositing electrodes to ensure good ohmic contact between the sample and the Au electrodes. To address the issue, after the EBL process, we implemented an additional Ar ion etching step to selectively remove the AlO_x layer at the electrode areas, thereby exposing the underlying MBT surface and ensuring good ohmic contact. This method offers two significant advantages. First, it effectively isolates the sample surface from direct contact with PMMA, mitigating damage during the fabrication. Second, the AlO_x layer acts as a protective layer, providing long-term preservation of the device. In previous studies on MBT device fabrication, no one has attempted to introduce an insulating AlO_x layer on the sample surface during the EBL process.

Compared to the method adopted by Yuanbo Zhang's group [*Science* 367, 895 (2020)], which uses an AlO_x-assisted exfoliation approach and shadow mask method, our method defines the Hall bar structures using EBL. The EBL method allows for fabricating much smaller structures, whereas shadow masks are limited to larger flakes. Moreover, in previous experiment, the AlO_x is positioned underneath the sample, leaving the top surface exposed. In contrast, our method involves depositing AlO_x on the top surface, which not only offers effective protection but also facilitates the potential integration of a top gate structure.

In the revised manuscript, we have included a detailed discussion of the distinctions between our new method and existing methods, highlighting the specific advantages of our technique.

2. It's hard to see the difference between before and after images in Fig. 1e. It may be better to show the line traces of the O_c signal before/after for flakes with and without AlO_x. Such a plot would be more transparent than Fig. 1f.

We thank the reviewer for this valuable suggestion. Indeed, our original presentation was not sufficiently clear. Following the reviewer’s suggestion, we added line traces in Fig. 1e and replotted the variation of O_c for selected spots in the revised Fig. 1f.

3. This study has not realized a deterministic fabrication of QAH effect. The authors should thus replace “resolve” to mitigate” in the abstract (line 33).

We thank the reviewer for pointing our wording issue. The word “resolve” has been replaced to “mitigate”.

4. It might be useful to generate a plot similar to Fig. 2g but for the coercive fields and/or the sharpness of the transition.

We thank the reviewer for the good suggestion. We have plotted the sharpness of σ_{xy} transition under magnetic fields, as marked by the blue squared dots. The trend in the sharpness of the plateau-plateau transitions aligns with the trend of the enhancement of the AH effect by AlO_x .

Fig. R8. Summarized $\Delta\sigma_{xy}/\Delta H$ values for the magnetic transition for 17 devices. Devices with AlO_x capping layer (blue) generally exhibit a sharper transition than those without AlO_x (red).

5. It might be helpful to compare similar data as in Fig. 4 for a lower-quality device without AlO_x . Such a comparison would strengthen the authors’ claim.

We thank the reviewer for the valuable suggestion. In the revised supplementary information, we added the similar plot for a lower-quality MBT without AlO_x . For the 7-SL device without AlO_x

capping layer, we found that the magnetic properties exhibited similar behaviors to those observed in the main figures. H_c shows V_g independent behaviors and T -dependent scaling analysis reveals similar critical exponent $\beta \sim 0.54$ ($\beta \sim 0.52$ for the device with AlO_x). However, the fitted $T_N \sim 20.0$ K is slightly lower ($T_N \sim 21.3$ K for the device with AlO_x), indicating weaker magnetism likely due to the absence of AlO_x . The deviation behavior at T below 8 K was also observed in prior neutron diffraction experiments [PRB, 101, 020412(R) (2020)], where the derived β in T range near T_N was found to be ~ 0.50 , consistent with our experiments. These consistent behaviors further support the claims made in the manuscript. The main results are shown in Fig. R9 below.

Fig. R9. Magnetic field dependent transport behaviors of a 7-SL MnBi_2Te_4 device without AlO_x at varied V_g and temperatures. **a-b**, μ_0H dependent σ_{xy} and σ_{xx} at the CNP ($V_g = 5$ V) for varied T s. **c**, H_c extracted from the field sweep data as a function of T . The black lines are the data fittings in the form of $(1-T/T_N)^\beta$. **d**, Colormap of σ_{xy} in the parameter space of μ_0H and V_g . The boundary between blue and orange region marks the V_g independent H_c .

Minor points:

-The acronym AH is not defined.

The acronym anomalous Hall (AH) was defined in the revised manuscript.

-Add references to “neither perfectly quantized [...] can be consistently reproduced.” line 69 on page 3

The relevant references were added.

Reviewer #4:

We thank the reviewer for the constructive feedback on our work. The one-to-one response to reviewer's questions are listed above. We hope that the revised manuscript meets the standards for publication in *Nature Communications*.

Below is a list of the main changes to the manuscript:

- (1) Title: no change
- (2) Author list: Xiaotian Yang, Wenbo Wang were added to the list.
- (3) Abstract: the word “resolve” in the abstract (line 33) was replaced by “mitigate”.
- (4) Main text: there are several changes to address the questions by reviewers, as discussed above in the response. The new contents are marked in red.
- (5) Figures and captions: we added line traces in Fig. 1e and plotted the variation of O_c for selected spots in the revised Fig. 1f. In Fig. 2g, we replotted the data based on the magnitude of σ_{xy} for the 17 devices, the device number (x -axis) was labeled by sorting the AH value from smallest to largest. We also summarized the variation of $\Delta\sigma_{xy}/\Delta H$ for the 17 devices on the right y -axis. In the inset of Fig. 2g, we added two histograms of σ_{xy} distribution with σ_{xy} on the x -axis and counts on the y -axis.
- (6) References: we added refs. 34, 35, 49 to address the issues raised by the reviewers.
- (7) Supplementary information: In section A, we included the result of optical contrast and atomic force microscopy measurements of MnBi_2Te_4 etched under different parameters. In section F, we added our preliminary MFM result measured on different areas of the same 7-SL MnBi_2Te_4 flake, and examined the effect of AlO_x capping layer on their magnetic properties. In section G, we incorporated the transport results for two MnBi_2Te_4 QAH devices with single-sided and double-sided AlO_x contacts.

Prepared by:

Chang Liu

Corresponding author

In this letter we provide a point-to-point response to the reviewers' comments.

In the following, the reviewers' original comments are shown by blue italic characters.

The authors' responses are shown by black normal characters.

Reviewer #1:

My previous questions have been addressed by the authors. I would recommend publication.

We thank the reviewer for the recommendation of our work.

Reviewer #2:

The authors have addressed all the questions with detailed and thoughtful responses. They have carefully revised both the text and the figures, incorporating our feedback. The addition of an MFM experiment not only enhances the clarity of the research but also demonstrates the authors' rigorous approach. I am now satisfied with the revised manuscript and fully support its publication.

We thank the reviewer for the nice summary and the favorable comments of our work.

Prepared by:

Chang Liu

Corresponding author